# Momentary subjective well-being depends on learning and not reward

**Bastien Blain[1]\*, Robb B Rutledge[1,2,3]\***

[1]Max Planck UCL Centre for Computational Psychiatry and Ageing Research, University College London, London, United Kingdom; [2]Wellcome Centre for Human Neuroimaging, University College London, London, United Kingdom; [3]Department of Psychology, Yale University, New Haven, United States

**Abstract** Subjective well-being or happiness is often associated with wealth. Recent studies suggest that momentary happiness is associated with reward prediction error, the difference between experienced and predicted reward, a key component of adaptive behaviour. We tested subjects in a reinforcement learning task in which reward size and probability were uncorrelated, allowing us to dissociate between the contributions of reward and learning to happiness. Using computational modelling, we found convergent evidence across stable and volatile learning tasks that happiness, like behaviour, is sensitive to learning-relevant variables (i.e. probability prediction error). Unlike behaviour, happiness is not sensitive to learning-irrelevant variables (i.e. reward prediction error). Increasing volatility reduces how many past trials influence behaviour but not happiness. Finally, depressive symptoms reduce happiness more in volatile than stable environments. Our results suggest that how we learn about our world may be more important for how we feel than the rewards we actually receive.

**\*For correspondence:**
b.blain@ucl.ac.uk (BB);
robb.rutledge@yale.edu (RBR)

**Competing interests:** The authors declare that no competing interests exist.

## Introduction

Decisions are guided by beliefs about states of the world. Some states are directly observable, like the potential prize for a bet. Other states, like the probability of winning, may not be directly observable but can be inferred from past events. Thus, learning from experience is essential for adaptive behaviour. In the standard theoretical framework, learning is driven by how unexpected the outcome is (i.e. by the prediction error): the difference between outcome and prediction (*Barto, 1995*). Sensitivity to the prediction error (i.e. the learning rate) flexibly adapts to environmental statistics. The decisions of both humans and non-human primates are consistent with a higher learning rate in more volatile environments, and subjects are more likely to stay on the same option after positive compared to negative feedback when reward probabilities change more frequently (*Behrens et al., 2007*; *Browning et al., 2015*; *Donahue and Lee, 2015*; *Massi et al., 2018*; *Mathys et al., 2011*).

Emotions are widely believed to play a role in adaptive behaviour (*Fredrickson, 2004*), but no computational framework exists to link them. Unexpected outcomes influence affective states, so that bad outcomes feel worse when unexpected than when expected, and good outcomes feel better when unexpected than when expected (*Mellers et al., 1997*; *Shepperd and Mcnulty, 2002*). It has recently been shown that reward expectations and reward prediction errors (RPEs), the difference between experienced and predicted rewards, can explain changes in affective state in the context of decision-making under uncertainty when learning is not required (*Rutledge et al., 2014*; *Rutledge et al., 2015*). A number of studies have found results consistent with the idea that happiness is modulated by past prediction errors (*Otto et al., 2016*) including a recent report showing in students that prediction errors due to exam performance influence real-world emotions (*Villano et al., 2020*). Mood has been proposed to represent environmental momentum, whether an

**eLife digest** Many people believe they would be happier if only they had more money. And events such as winning the lottery or receiving a large pay rise do make people happy, at least temporarily. But recent studies suggest that the main factor driving happiness on such occasions is not the size of the reward received. Instead, it is how well that reward matches up with expectations. Receiving a 10% pay rise when you were expecting 1% will make you feel happier than receiving 10% when you had been expecting 20%.

This difference between an expected and an actual reward is referred to as a reward prediction error. Reward prediction errors have a key role in learning. They motivate people to repeat behaviours that led to unexpectedly large rewards. But they also enable people to update their beliefs about the world, which is rewarding in itself. Could it be that reward prediction errors are associated with happiness mainly because they help us understand the world a little better than before?

To test this idea, Blain and Rutledge designed a task in which the likelihood of receiving a reward was unrelated to the size of the reward. This study design makes it possible to separate out the contributions of learning versus reward to moment-by-moment happiness.

In the task, volunteers had to decide which of two cars would win a race. In the 'stable' condition, one of the cars always had an 80% chance of winning. In the 'volatile' condition, one car had an 80% chance of winning for the first 20 trials. The other car then had an 80% chance of winning for the next 20 trials. The volunteers were not told these probabilities in advance, but had to work them out by playing the game. However, on every trial, the volunteers were shown the reward they would receive if they chose either of the cars and that car went on to win. The size of the rewards varied at random and was unrelated to the likelihood of a car winning.

Every few trials, the volunteers were asked to indicate their current level of happiness on a scale. The results showed that volunteers were happier after winning than after losing. On average they were also happier in the stable condition than in the volatile condition. This was especially true for volunteers with pre-existing symptoms of depression. Moreover, happiness after wins did not depend on how large the reward they got was, but instead simply on how surprised they were to win.

These results suggest that how we learn about the world around us can be more important for how we feel than rewards we receive directly. Measuring happiness in various types of environment could help us understand factors affecting mental health. The current results suggest, for example, that uncertain environments may be especially unpleasant for people with depression. Further research is needed to understand why this might be the case. In the real world, rewards are often uncertain and infrequent, but learning may nevertheless have the potential to boost happiness.

environment is getting better or worse, which could be a useful variable for adaptive behaviour (*Eldar et al., 2016*; *Eldar and Niv, 2015*).

Impairments in reward and emotion processing are associated with depression (*Shepperd and Mcnulty, 2002*). Learning in depression has been extensively studied, but there is not consistent evidence for a specific deficit (*Blanco et al., 2013*; *Cella et al., 2010*; *Chase et al., 2010*; *Gillan et al., 2016*; *Herzallah et al., 2013*; *Kunisato et al., 2012*; *Mueller et al., 2015*; *Pechtel et al., 2013*; *Robinson et al., 2012*; *Taylor Tavares et al., 2008*; *Thoma et al., 2015*; *Vrieze et al., 2013*), as reviewed in *Huys et al., 2013*; *Scholl and Klein-Flügge, 2018*. The ability of individuals to appropriately adjust learning rates to environmental volatility is associated with anxiety symptoms (*Browning et al., 2015*). Individuals with high trait anxiety show reduced ability to adjust updating of outcome expectancies for aversive outcomes to the volatility of the environment compared to individuals with low trait anxiety. Failure to appropriately adjust learning to environmental volatility has not been established in depression. Affective states reflect subjective estimates of uncertainty, which predict the dynamics of subjective and physiological stress responses (*de Berker et al., 2016*). Overall mood during risky decision-making tasks is reduced with increasing depression severity, both in the laboratory and using remote smartphone-based data collection (*Rutledge et al., 2017*).

Here, our goal was to quantify the relationship between mood and adaptive behaviour in two common reinforcement learning tasks (*Figure 1*): one in which reward probabilities do not change (stable) and one in which reward probabilities periodically change (volatile). We addressed the following questions: (1) Is mood more sensitive to learning-relevant or learning-irrelevant variables in established reinforcement learning models? (2) Do mood dynamics adjust to environmental volatility? (3) Are mood dynamics affected by depression in the context of learning?

## Results

To investigate the relationship between mood dynamics and adaptive behaviour, we adapted a task design employed in previous studies (*Behrens et al., 2007*; *Massi et al., 2018*). Participants repeatedly made decisions between two cars racing against each other (corresponding to a one-armed bandit because only one car can win each race). The reward magnitude associated with each car was explicit and assigned randomly on each trial, whereas the outcome probability was implicit and had to be inferred from the outcomes of previous trials. Participants (n = 75) completed two learning tasks with a break in between tasks (counterbalanced across participants). In the stable environment, the 'best' car won 80% of the races on average. In the volatile environment, the 'best' car had an 80% probability of winning and the 'best' car switched every 20 trials (see *Figure 1*). Participants were asked to report their current happiness after every three to four trials. Unlike in previous studies in humans using a similar task design (*Behrens et al., 2007*; *Browning et al., 2015*), participants were informed that they would complete stable and volatile tasks. Which environment they were in was indicated at the start of each task. Subjects were given no guidance as to how they should use this information, although we expected this manipulation to increase the difference in behavioural sensitivity between environments relative to previous studies.

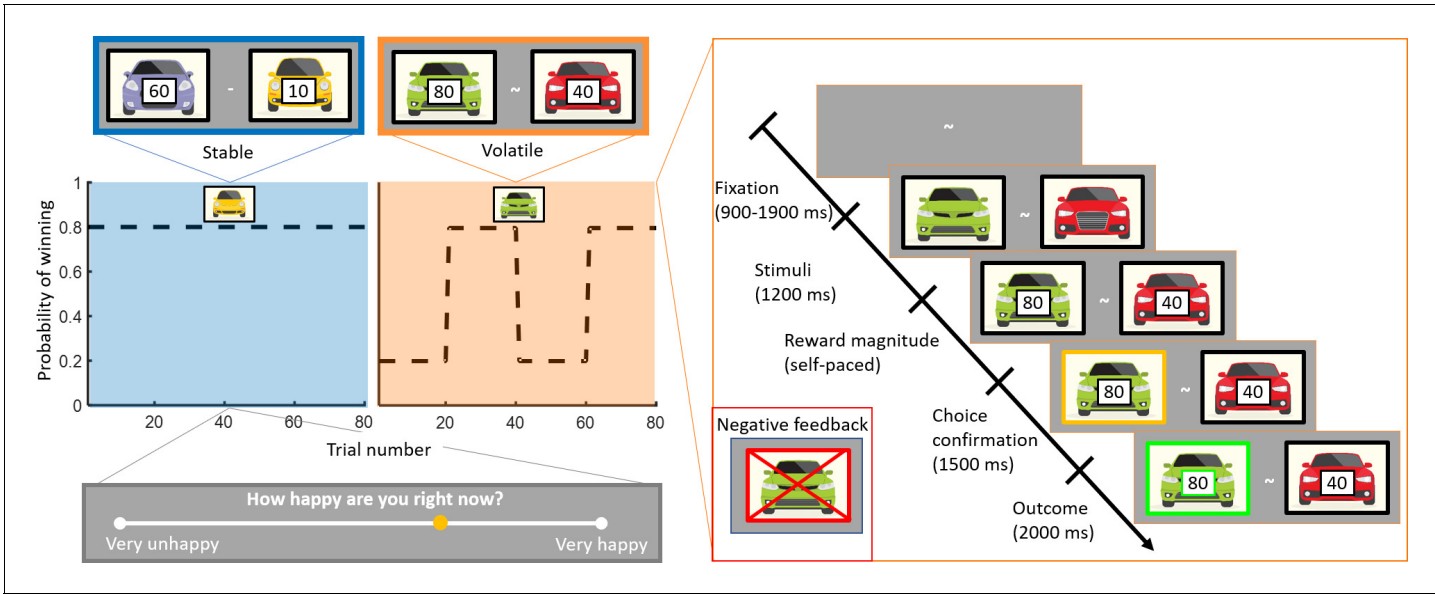

**Figure 1.** Experimental design. Subjects (n = 75) performed a one-armed bandit reinforcement learning task, choosing repeatedly between two cars. They were instructed to maximise their cumulative points. In the stable task (80 trials), the probability to win for the best car was 80%. In the volatile task (80 trials), reward probabilities switched between 80% for one car and 80% for the other car every 20 trials. Task order was counterbalanced across subjects (see Materials and methods). The reward available for each car was randomly determined on each trial and unrelated to the probability of winning. Every three to four trials, subjects were asked to report 'How happy are you right now?' by moving a cursor on a line. Each trial started with a fixation symbol in the centre of the screen. Then, the stimuli were displayed but choice was not permitted. The potential reward for each car was then displayed and participants were free to choose an option without any time constraints. The chosen option was outlined by a yellow frame. Finally, the outcome was displayed. Both the car and the reward magnitude frames were green if the chosen car won the race (example shown). The car frame was red and crossed out if the chosen car lost (example shown in inset).

## Learning rates change with environmental volatility

Participants chose the option with the higher expected value (i.e. choice accuracy) more often than chance (see *Figure 2A*) in the stable environment (82.1 ± 1.1% (mean ± SEM), z = 7.5, p < $10^{-13}$) and in the volatile environment (73.3 ± 1.0%, z = 7.5, p < $10^{-13}$). Participants chose the higher probability option more often than chance in the stable environment (80.8 ± 1.0%, mean ± SEM, z = 7.5, p < $10^{-13}$) and in the volatile environment (62.6 ± 1.1%, z = 7.2, p < $10^{-12}$). To ensure that participants incorporated information about the magnitudes of potential rewards into their decisions, we considered only trials where the car with the lower outcome probability had the higher expected value. Subjects chose the low probability car in these trials more often than chance (stable: 61.2 ± 3.0%, z = 3.5, p < 0.001; volatile: 75.6 ± 2.7%, z = 6.4, p < $10^{-9}$). These results are consistent with participants integrating both probability and reward to make their decisions.

Multiplicative and additive models that integrate probability and reward in different ways have been widely used to explain behaviour across stable and volatile environments (*Behrens et al., 2007*; *Browning et al., 2015*; *Donahue and Lee, 2015*; *Massi et al., 2018*). Both types of models include the same learning component for updating the probability estimate for each car to win (*Equations 1 and 2*), based on the probability prediction error (PPE), the difference between the outcome (0 or 1) and the estimated probability of winning:

$$P_{car1\ wins}(t+1) = P_{car1\ wins}(t) + \alpha\ PPE(t), \tag{1}$$

$$PPE(t) = Outcome(t) - P_{car1\ wins}(t), \tag{2}$$

where $P_{car1\ wins}(t)$ is the estimated probability for car 1 winning on trial t, and $\alpha$ is the learning rate. For choices to car 2, a similar equation applies. Multiplicative and additive models differ regarding implementation of choice predictions with probability and reward magnitude either integrated multiplicatively (*Behrens et al., 2007*; *Browning et al., 2015*) or additively (*Donahue and Lee, 2015*; *Farashahi et al., 2017*; *Massi et al., 2018*).

The multiplicative selector resembles the maximisation of expected utility common to economic decision models (*Kahneman and Tversky, 1979*):

$$EU_{car1}(t) = \max[\min[\eta((P_{car1wins}(t) - 0.5) + 0.5), 1], 0] \times R_{car1wins}(t) \tag{3}$$

$$P_{car1\ chosen}(t) = \frac{1}{1 + \exp(-\beta(EU_{car1}(t) - EU_{car2}(t)))}, \tag{4}$$

where $\eta$ is a free parameter related to the level of risk aversion and $\beta$ is the inverse temperature (i.e. choice stochasticity or precision), and $R_{car1wins}(t)$ is the reward magnitude if car 1 is chosen and wins. The multiplicative model captured choice data in both stable (pseudo-$r^2$ = 0.54 ± 0.02, mean ± SEM) and volatile environments (pseudo-$r^2$ = 0.41 ± 0.02).

In contrast, the additive selector is implemented as follows:

$$P_{car1chosen}(t) = \frac{1}{1 + \exp(-\beta(\phi\ \Delta Probability(t) + (1-\phi)\Delta Reward(t)))} \tag{5}$$

where $\Delta$Probability corresponds to the difference in estimated probability between the options and $\Delta$Reward corresponds to the difference in normalised reward magnitude between the options, and $\phi$ represents the relative weight of probability and magnitude on choice. The additive model captured choice data in the stable (pseudo-$r^2$ = 0.62 ± 0.02) and volatile environments (pseudo-$r^2$ = 0.45 ± 0.02, see *Figure 2A*).

Model comparison demonstrated that the additive model better explained choice data with the same number of parameters as the multiplicative model in both stable ($\Delta$BIC = 690) and volatile environment ($\Delta$BIC = 321, see *Table 1*). Our results are consistent with similar findings obtained in highly trained non-human primates using a task design in which changes between stable and volatile environments were signalled (*Massi et al., 2018*) as in the present study.

We next examined whether model fits were consistent with subjects integrating both potential reward magnitudes and probabilities to make decisions. The relative weight of probability and reward magnitude in the additive model ($\phi$) was balanced on average (stable: $\phi$ = 0.57 ± 0.017

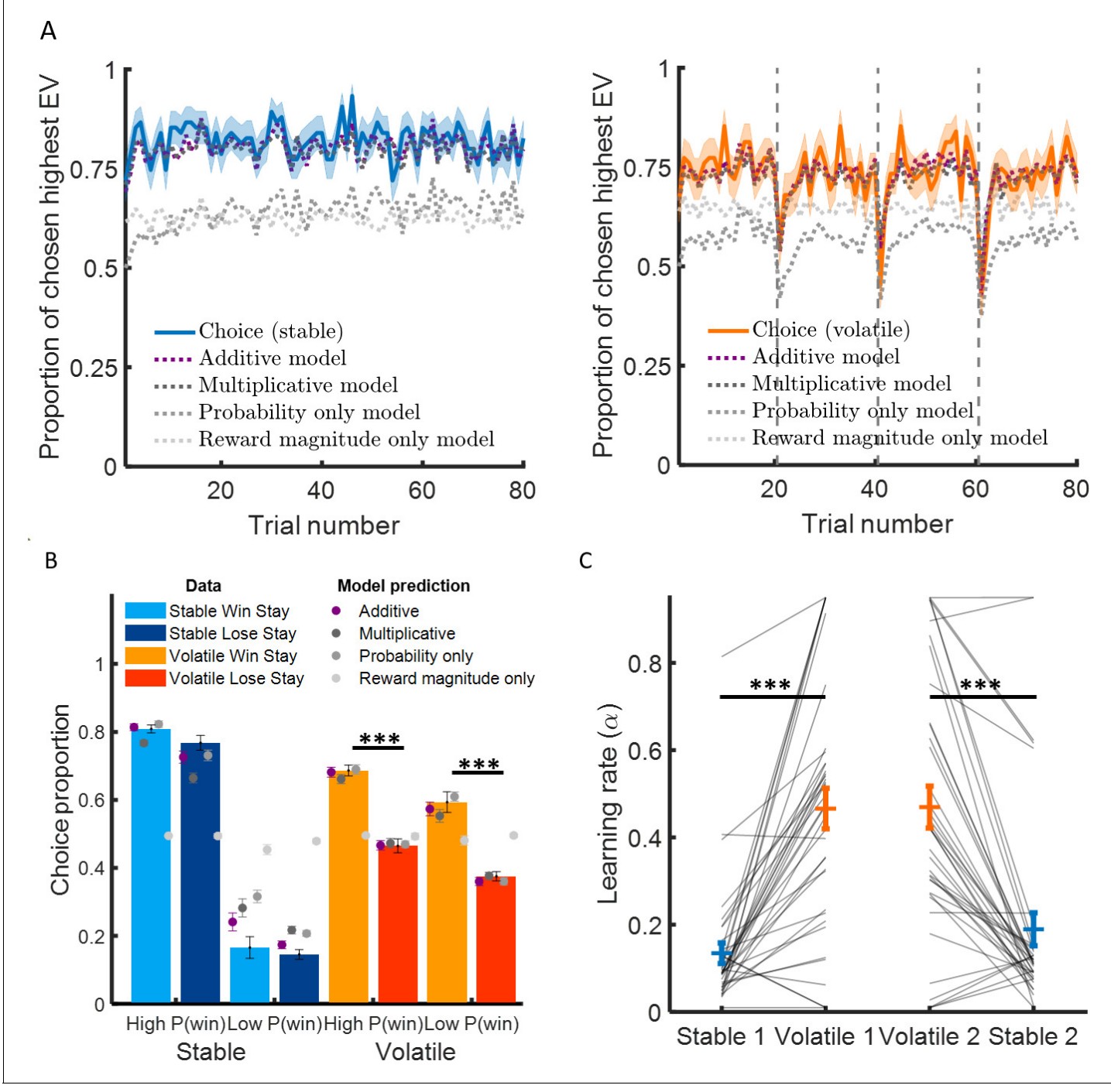

**Figure 2.** Learning rate adapts to environmental volatility. (**A**) Participants chose the option with the highest expected value 82% of the time in the stable environment (blue curve, left panel) and 73% of the time in the volatile environment (orange curve, right panel). The additive model containing three parameters (a learning rate determining the sensitivity to prediction error, an inverse temperature reflecting choice stochasticity, and a relative weight for probability and reward magnitude in choice) fitted choice data well (black dashed lines) in the stable environment (mean pseudo-$r^2$ = 0.62) and the volatile environment (mean pseudo-$r^2$ = 0.45). (**B**) Participants chose more often the option with the higher probability in the stable environment compared to the volatile environment. Critically, participants stayed on the same option more often if choosing that option resulted in the car winning (light orange) compared to the car losing (dark orange) in the volatile environment compared to the stable environment (light blue and dark blue represent staying after winning and losing, respectively). This suggests that participant behaviour was more sensitive to feedback in the volatile than stable environment, as an agent with a higher learning rate would be. Additive model predictions show a similar difference in feedback sensitivity across environments (purple). (**C**) Learning rates were higher in the volatile environment (orange) compared to the stable environment (blue).

*Figure 2 continued on next page*

*Figure 2 continued*

This was true for participants completed the stable learning task before (stable 1) or after (stable 2) the volatile learning task. Error bars represent SEM. *p < 0.05, ***p < 0.001.

[mean ± SEM]; volatile: $\phi$ = 0.44 ± 0.027), suggesting that subjects integrated both probabilities and reward magnitudes to make decisions. Omitting $\phi$ and evaluating simpler models that considered only probabilities ($\alpha$ and $\beta$) or reward magnitudes ($\beta$ only) resulted in worse fits (see *Table 1*). Lower BIC for additive and multiplicate models compared to the simpler models confirmed that subjects considered both probability and reward magnitude when making decisions.

We then asked whether greater environmental volatility was associated with higher learning rates as observed in previous studies (*Behrens et al., 2007*; *Browning et al., 2015*; *Massi et al., 2018*). A simple prediction for standard reinforcement learning models is that subjects should stay more on the same option after winning than losing (*Figure 2B*). Subjects did not stay more on the same option after winning than losing for the stable environment (difference in choice proportion, high probability car: 4.1 ± 1.9%, mean ± SEM, z = 1.7, p = 0.098; low probability car: 1.3 ± 3.2%, z = -0.43, p = 0.67). Subjects stayed on the same option more after winning than losing in the volatile environment (difference in choice proportion, high probability car: 22.1 ± 2.3%, z = 6.6, p < $10^{-10}$, low probability car: 21.4 ± 3.7%, z = 4.8, p < $10^{-5}$). Subjects stayed on the same option after winning compared to losing more in volatile than stable environments (difference volatile – stable, high probability car: 18.1 ± 2.7%, z = 5.7, p < $10^{-7}$, low probability car: 19.8 ± 4.1%, z = 4.0, p < $10^{-4}$; see *Figure 2B*).

We then checked that the predictions generated by a reinforcement learning model fit separately to each environment correspond to observed behavioural patterns described above in model-independent analyses (predicted difference in choice proportion after winning and losing in stable [high probability car: 8.8 ± 1.6%, z = 5.2, p < $10^{-6}$, low probability car: 6.1 ± 2.6%, z = 1.5, p = 0.12] and volatile environments [high probability car: 21.5 ± 2.2%, z = 6.9, p < $10^{-11}$, low probability car: 21.1 ± 2.7%, z = 5.9, p < $10^{-8}$; see *Figure 2B*]). The model predictions were able to capture observed differences in behaviour following wins and losses and also the difference in in choice proportion after winning and losing between volatile and stable environments (high probability car: 12.8 ± 2.4%, z = 4.6, p < $10^{-5}$; low probability car: 14.3 ± 3.6, z = 3.4, p < 0.001). We found that learning rates (*Figure 2C*) were substantially higher in volatile than stable environments (stable $\alpha$ = 0.16 ± 0.02, mean ± SEM; volatile $\alpha$ = 0.47 ± 0.03; difference volatile – stable: 0.31 ± 0.03, z = 6.9, p < $10^{-11}$). Overall, these results demonstrate that the learning rate increases substantially in the volatile compared to the stable environment, in line with previous studies (*Behrens et al., 2007*; *Browning et al., 2015*; *Massi et al., 2018*).

**Table 1.** Choice model comparison results.

The 'Additive' model refers to a model implementing a weighted sum of probability difference and reward magnitude difference when making decisions (*Donahue and Lee, 2015*; *Farashahi et al., 2019*; *Farashahi et al., 2017*; *Massi et al., 2018*; *Rouault et al., 2019*). The 'Multiplicative' model refers to the model first used to describe behaviour in this task, which integrates reward and probability information multiplicatively (*Behrens et al., 2007*; *Browning et al., 2015*). The 'Probability only' model includes only the probability component of the additive model and the 'Magnitude only' model includes only the magnitude component of the additive model. ΔBIC refers to the Bayesian Information Criterion computed for each model compared to the additive model, the preferred model in both stable and volatile environments.

| Model | Number of parameters | Stable pseudo-$r^2$ | Volatile pseudo-$r^2$ | Stable BIC | Volatile BIC | Stable ΔBIC | Volatile ΔBIC |
|---|---|---|---|---|---|---|---|
| Additive | 3 | 0.62 | 0.45 | 4134 | 5605 | 0 | 0 |
| Multiplicative | 3 | 0.54 | 0.41 | 4824 | 5926 | 690 | 321 |
| Probability only | 2 | 0.35 | 0.16 | 6104 | 7635 | 1970 | 2030 |
| Magnitude only | 1 | 0.14 | 0.23 | 7473 | 6739 | 3338 | 1134 |

## Happiness is more sensitive to learning-relevant than learning-irrelevant variables

We next examined how happiness changes over time during the tasks. Subjects varied their happiness ratings in both the stable (SD = 24.2 ± 1.1, mean ± SEM) and volatile (SD = 25.0 ± 1.1) environments. They were happier on average after winning than after losing (stable: 63.8 ± 1.9 vs 34.5 ± 2.0, z = 7.5, p < 10$^{-13}$; volatile: 61.9 ± 1.8 vs 33.6 ± 2.0, z = 7.5, p < 10$^{-13}$, *Figure 3A*). Participants were happier on average in the stable environment than in the volatile environment (stable: 55.0 ± 1.7, volatile: 49.5 ± 1.6, z = 3.7, p < 0.001). This effect may be at least partly due to lower choice accuracy in volatile environments, and the difference in average happiness between environments was correlated between participants with the difference in choice accuracy in terms of EV maximisation (Spearman's ρ(73) = 0.24, p < 0.05).

Previous studies have reported that momentary happiness in response to outcomes in a probabilistic reward task were explained by recent RPEs when maximising cumulative reward does not require learning (*Rutledge et al., 2014*; *Rutledge et al., 2015*). In our task, maximising cumulative reward requires learning the outcome probability. In this context, PPEs (depending on whether the outcome was a win or a loss and the subjective probability of that outcome) are relevant to learning but RPEs (depending on the magnitude of the reward received and the expected value of the chosen option) are not relevant to learning or future behaviour. Reward information was choice relevant and choices were driven by the (additive) integration of the estimated outcome probability and the magnitude of potential rewards. Therefore, we tested whether happiness was more strongly associated with the PPEs used for learning or alternatively by RPEs that incorporate learning-irrelevant reward magnitudes. We compared two models which both use the subjective probability as estimated in the additive choice model to compute prediction errors:

$$Happiness(t) = w_0 + w_{\widehat{PPE}} \sum_{j=1}^{t} \gamma^{t-j} \widehat{PPE}_j, \tag{6}$$

where $\widehat{PPE}$ refers to the probability prediction error (PPE), defined as the difference between the outcome (one for win, 0 for loss) and the subjective probability estimated from the additive choice model, $w_0$ is a constant term, $w_{PPE}$ is a weight capturing the influence of past PPEs, and $0 \leq \gamma \leq 1$ is a forgetting factor that makes events in more recent trials more influential than those in earlier trials;

$$Happiness(t) = w_0 + w_{\widehat{RPE}} \sum_{j=1}^{t} \gamma^{t-j} \widehat{RPE}_j, \tag{7}$$

where $\widehat{RPE}$ is the difference between reward magnitude and the expected value of the chosen option computed based on the subjective probability estimated from the additive choice model. Reward magnitudes were rescaled from 0 to 1.

Mood fluctuations were better explained by a model including past PPEs than by a model including past RPEs, both in the stable (BIC$_{PPE}$ = −698, BIC$_{RPE}$ = −299, ΔBIC = 399) and volatile (BIC$_{PPE}$ = −559, BIC$_{RPE}$ = −319, ΔBIC = 240) environments (see *Table 2* and *Figure 4A*). This result holds for a broader model space including other definitions of the prediction error terms (see *Table 2* and *Figure 4*):

$$Happiness(t) = w_0 + w_{PPE} \sum_{j=1}^{t} \gamma^{t-j} PPE_j, \tag{8}$$

where PPE refers to the objective PPE defined as the difference between the outcome sign and the objective probability of the chosen option (0.2 or 0.8), and also:

$$Happiness(t) = w_0 + w_{RPE} \sum_{j=1}^{t} \gamma^{t-j} RPE_j, \tag{9}$$

where RPE is computed by taking the difference between the reward magnitude and the objective expected value (potential reward multiplied by the objective probability of the chosen option) as above.

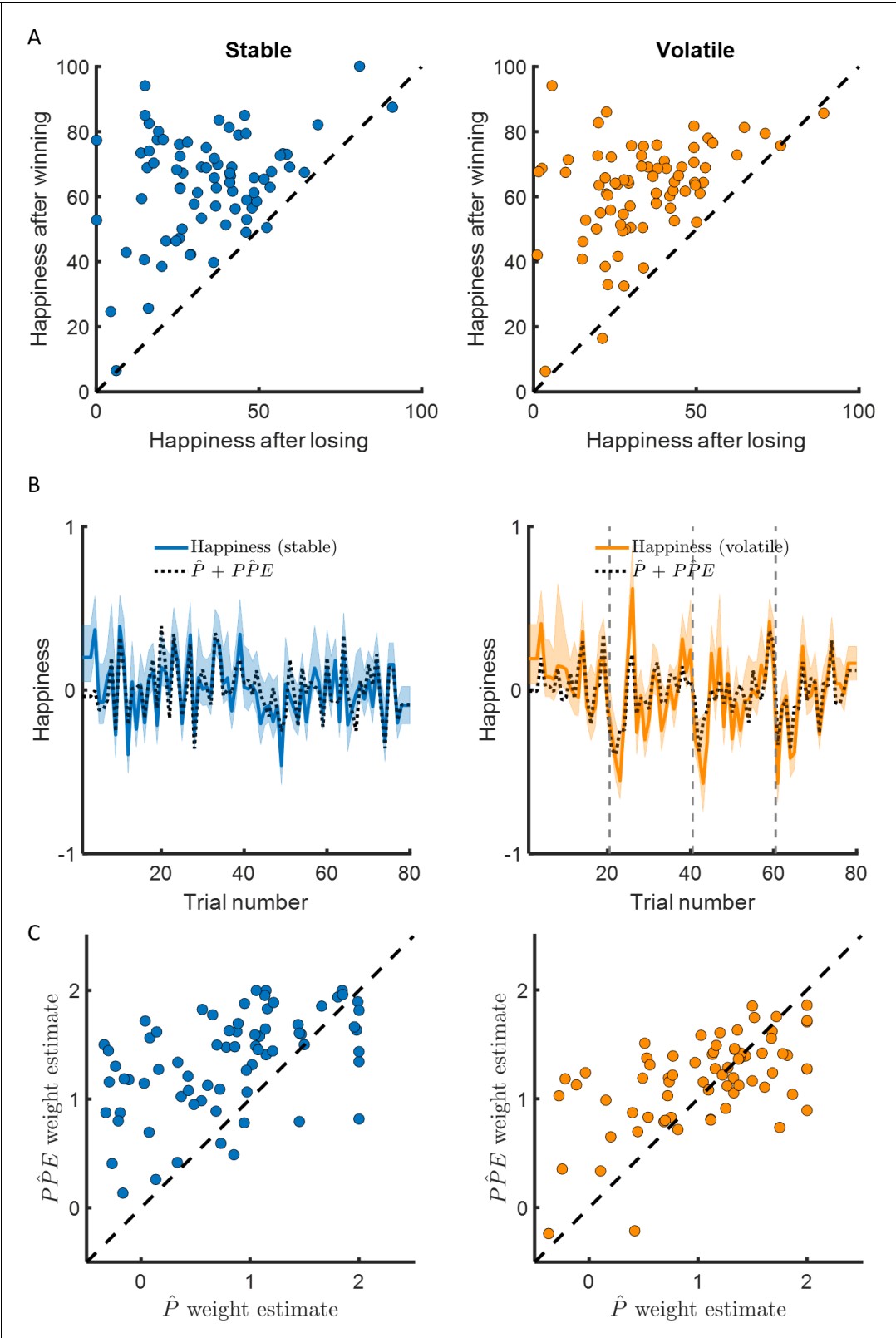

**Figure 3.** Happiness is associated with probability and probability prediction error. (**A**) Most participants were happier when their chosen car won compared to when their chosen car lost (97% of participants in the stable environment, 96% in the volatile environment, in the left and right panel, respectively). (**B**) Momentary happiness was best explained by a model (black dotted lines) including both the chosen probability estimate and the probability prediction error (PPE) derived from the additive choice model in addition to a forgetting factor and a baseline mood parameter, for both

*Figure 3 continued on next page*

*Figure 3 continued*

the stable (mean $r^2$ = 0.58) and the volatile (mean $r^2$ = 0.62) environments. Happiness ratings were z-scored for individual participants before model fitting. The shaded areas represent SEM. (C) The chosen probability (denoted P) and the PPE parameters were significantly different from 0 for both environments. Both variables are significantly associated with changes in affective state over time. PPE weight was significantly higher than P weight in both the stable and volatile environments. See *Figure 3—figure supplement 1* related to the win loss model parameters.
The online version of this article includes the following figure supplement(s) for figure 3:

**Figure supplement 1.** Loss weights on happiness are greater than win weights.

We then tested whether mood fluctuations were additionally sensitive to current expectations, as shown in risky choice tasks that do not require learning (*Rutledge et al., 2014*; *Rutledge et al., 2015*). Again, two types of expectations may explain mood fluctuations: a subjective probability relevant to learning, or expected values that incorporate learning-irrelevant reward magnitudes. We compared the following models:

$$Happiness(t) = w_0 + w_{\hat{P}} \sum_{j=1}^{t} \gamma^{t-j} (\hat{P} - 0.5) + w_{\widehat{PPE}} \sum_{j=1}^{t} \gamma^{t-j} \widehat{PPE}_j, \tag{10}$$

where $P$ is the probability estimated with the additive choice model, and:

$$Happiness(t) = w_0 + w_{\widehat{EV}} \sum_{j=1}^{t} \gamma^{t-j} \left( \widehat{EV} - \bar{EV} \right) + w_{\widehat{PPE}} \sum_{j=1}^{t} \gamma^{t-j} \widehat{PPE}_j, \tag{11}$$

where $\widehat{EV}$ is the product between $\hat{P}$ and the reward magnitude and this term is mean-centred.

The model including choice probability better explained happiness ratings (stable: mean $r^2$ = 0.58; volatile: mean $r^2$ = 0.62) than the model including the expected value in the stable ($BIC_{\hat{P}+\widehat{PPE}}$ = −882, $BIC_{EV+PPE}$ = −752, ΔBIC = 130) and in the volatile ($BIC_{\hat{P}+PPE}$ = −1147, $BIC_{EV+PPE}$ = −691, ΔBIC = 454) environments (see *Table 2* and *Figure 4B*). The probability and PPE weights were significantly different from 0 at the group level in both the stable ($w_{\hat{P}}$ = 0.74 ± 0.09, z = 6.2,

**Table 2.** Happiness model comparison results.
PPE is probability prediction error, RPE is reward prediction error, P is the probability estimate, EV is the expected value, R is reward, $\bar{R}$ is the reward average, and RP is a free parameter corresponding to the reference point above and below which happiness would increase or decrease. The hat over a variable indicates that it incorporates trial-by-trial choice probability estimated from the additive choice model. ΔBIC refers to the comparison of the model scores using the Bayesian Information Criterion (BIC) compared to the $\hat{P} + \widehat{PPE}$ model. Happiness ratings were z-scored within individuals and all models included a constant term and a forgetting factor γ in addition to the parameters indicated.

| Model | Number of parameters | Stable mean $r^2$ | Volatile mean $r^2$ | Stable BIC | Volatile BIC | Stable ΔBIC | Volatile ΔBIC |
|---|---|---|---|---|---|---|---|
| $\widehat{PPE}$ | 3 | 0.50 | 0.44 | −698 | −559 | 184 | 587 |
| $\widehat{RPE}$ | 3 | 0.38 | 0.39 | −299 | −319 | 583 | 826 |
| PPE | 3 | 0.48 | 0.42 | −640 | −436 | 242 | 710 |
| RPE | 3 | 0.36 | 0.36 | −223 | −212 | 658 | 934 |
| $\hat{P} + \widehat{PPE}$ | 4 | 0.58 | 0.62 | −882 | −1146 | 0 | 0 |
| $\widehat{EV} + \widehat{PPE}$ | 4 | 0.56 | 0.53 | −752 | −691 | 130 | 454 |
| $\widehat{EV} + \widehat{RPE}$ | 4 | 0.43 | 0.48 | −224 | −370 | 657 | 775 |
| $\hat{P} + \widehat{RPE}$ | 4 | 0.47 | 0.51 | −370 | −504 | 511 | 641 |
| $R - \bar{R}$ | 3 | 0.36 | 0.44 | −242 | −471 | 640 | 675 |
| $R - RP$ | 4 | 0.35 | 0.42 | 47 | −193 | 929 | 952 |
| Win −loss | 4 | 0.57 | 0.63 | −848 | −1181 | 34 | −35 |

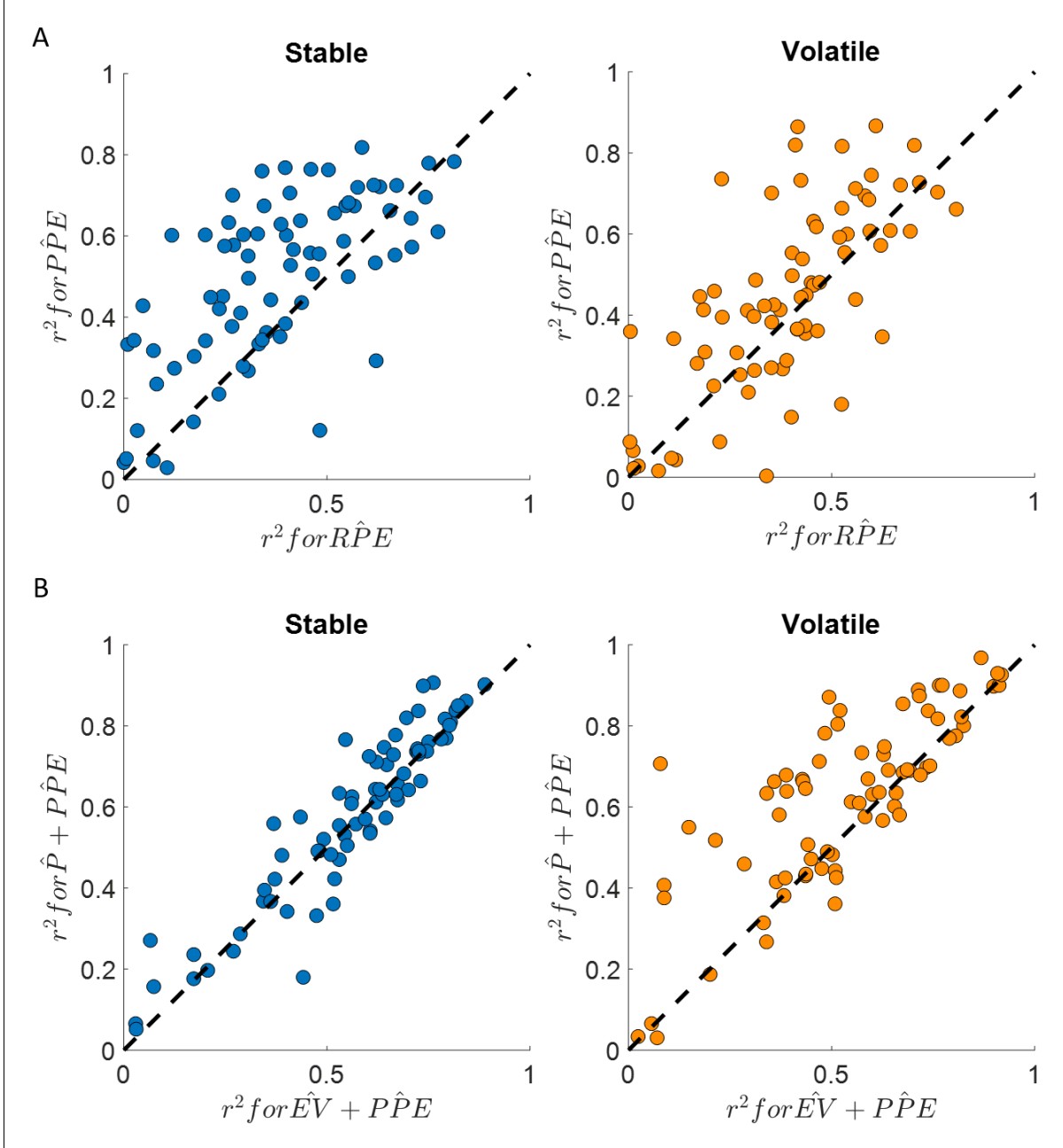

**Figure 4.** Happiness is more strongly associated with learning than choice. (**A**) Comparison between the $r^2$ for the happiness model including a PPE term (denoted $\widehat{PPE}$) estimated in the additive choice model (y axis) and the $r^2$ for the happiness model including an RPE term instead (denoted $\widehat{RPE}$). Both models had the same number of parameters. The $\widehat{PPE}$ model accounted for more variance in mood ratings on average in both stable (blue) and volatile (orange) learning tasks. Dots above the dashed line correspond to subjects for whom more variance in happiness is explained by the $\widehat{PPE}$ compared to the $\widehat{RPE}$ model. (**B**) The $\widehat{PPE}$ model including the chosen estimated probability (denoted $\hat{P}$ and estimated from the additive choice model) better explained happiness ratings than a $\widehat{PPE}$ model including expected value (denoted $\widehat{EV}$) for both the stable (blue) and volatile (orange) environments with both models having the same number of parameters. Dots above the dashed line correspond to subjects where more variance in happiness is explained by the $\hat{P} + \widehat{PPE}$ compared to the $\widehat{EV} + \widehat{PPE}$ model. See **Figure 4—figure supplement 1** for the estimated model frequency or each model and **Table 2** for other model comparison metrics.

The online version of this article includes the following figure supplement(s) for figure 4:

**Figure supplement 1.** Estimated model frequency.

$p < 10^{-9}$; $w_{PPE} = 1.32 \pm 0.06$, $z = 7.5$, $p < 10^{-14}$) and the volatile environments ($w_{P} = 0.94 \pm 0.09$, $z = 6.7$, $p < 10^{-10}$; $w_{PPE} = 1.14 \pm 0.05$, $p < 10^{-14}$, **Figure 3C**).

We next extended the model space with plausible alternative models. We included two models incorporating the history of reward magnitude. In the first model, we centred the reward magnitude regressor for each participant. This model thus predicts that reward magnitudes larger than the averaged reward magnitude will increase happiness, and that the larger the reward magnitude, the greater the happiness.

$$Happiness(t) = w_0 + w_R \sum_{j=1}^{t} \gamma^{t-j} \left( R_j - \bar{R} \right),$$ (12)

where $R_j$ is the reward magnitude at trial $j$ and $\bar{R}$ is the average reward magnitude. Instead of assuming a reference point of the average reward, we also used a free parameter in a subsequent model above and below which reward magnitudes increase or decrease happiness, respectively:

$$Happiness(t) = w_0 + w_R \sum_{j=1}^{t} \gamma^{t-j} \left( R_j - RP \right),$$ (13)

where $RP$ is a free parameter corresponding to the reference point in an individual subject. If this value is 0, receipt of rewards always increases happiness in proportion to the reward magnitude, so this model also provides a test of whether failing to obtain reward decreases happiness during reinforcement learning. The average reward magnitude was $25.5 \pm 2.5$ (mean $\pm$ SD) points in the stable environment and $22.6 \pm 3.0$ points in the volatile environment. The reference point $RP$ estimated in model 14 was on average $14.2 \pm 9.5$ points and greater than 0 in the stable environment ($z = 7.0$, $p < 10^{-11}$) and $14.8 \pm 7.3$ points and greater than 0 in the volatile environment ($z = 7.4$, $p < 10^{-12}$). This result supports the idea that obtaining 0 points on a trial is aversive: failing to obtain reward decreases happiness in our tasks. We also included two additional models incorporating $\widehat{RPE}$.

$$Happiness(t) = w_0 + w_{\widehat{EV}} \sum_{j=1}^{t} \gamma^{t-j} \left( \widehat{EV}_j - \bar{EV} \right) + w_{\widehat{RPE}} \sum_{j=1}^{t} \gamma^{t-j} \widehat{RPE}_j,$$ (14)

where $\widehat{EV}$ corresponds to the expected value of the chosen option, corresponding to the weighted sum of probability and reward estimated based on each individual participant's choices with the additive choice model, $\bar{EV}$ corresponds to the averaged expected reward, and $\widehat{RPE}$ corresponds to the difference between the outcome reward magnitude and $\widehat{EV}$. We also included a model combining the estimated probability with the reward prediction error.

$$Happiness(t) = w_0 + w_{\hat{P}} \sum_{j=1}^{t} \gamma^{t-j} \left( \hat{P}_j - 0.5 \right) + w_{\widehat{RPE}} \sum_{j=1}^{t} \gamma^{t-j} \widehat{RPE}_j,$$ (15)

Besides the constant and the forgetting factor, the $\hat{P} + \widehat{PPE}$ model includes two parameters, one for predictions ($w_P$) and one for probability prediction error ($w_{PPE}$).

We also asked whether the history of wins (excluding any information about reward magnitude) and losses could account for happiness by fitting the following model:

$$Happiness(t) = w_0 + w_{win} \sum_{j=1}^{t} \gamma^{t-j} win_j - w_{loss} \sum_{j=1}^{t} \gamma^{t-j} loss_j$$ (16)

As reported in **Table 2**, the model evidence for this new model was similar to the $\hat{P} + \widehat{PPE}$ model overall. We next used estimated model frequency to compare both models. The $\hat{P} + \widehat{PPE}$ is preferred to the win-loss model in the stable environment ($EF_{P+PPE} = 0.65 \pm 0.05$, $EF_{win-loss} = 0.35 \pm 0.05$, exceedance probability = 0.99). However, both models performed similarly in the volatile environment ($EF_{P+PPE} = 0.50 \pm 0.06$, $EF_{win-loss} = 0.50 \pm 0.06$, exceedance probability = 0.48; see **Figure 4— figure supplement 1**).

We next asked whether an alternative analysis could test whether happiness was influenced by trial-by-trial probability estimates, the key difference between the models. If the weights for the two

terms of the $\hat{P} + \widehat{PPE}$ model are identical, the equation mathematically reduces to a constant plus an exponentially weighted average of previous wins. However, the $w_{PPE}$ parameter was larger than $w_{P^-}$ in both stable ($z = 6.0$, $p < 10^{-8}$) and volatile tasks ($z = 2.35$, $p < 0.05$). The difference between $w_{PPE}$ and $w_{P^-}$ was significantly larger in the stable compared to the volatile task ($z = 3.7$, $p < 0.001$). Comparison of weights across tasks therefore suggests a reduced impact of expectations on happiness as environmental volatility increases. We next computed the residuals of the win-loss model (which does not include probability estimates) and tested for a correlation with trial-by-trial probability estimates. Because prediction errors are equal to outcomes minus expectations and numerically $w_{P^-}$ is lower than $w_{PPE}$ in both environments, the overall influence of probability on happiness should be negative after accounting for the impact of wins and losses. In the stable environment, we found the expected negative correlation between the win-loss model residuals and trial-by-trial probability estimates (average Spearman's $\rho(73) = -0.06 \pm 0.03$, $z = 2.2$, $p = 0.03$). This relationship was not present in the volatile environment (average Spearman's $\rho(73) = -0.02 \pm 0.03$, $z = 0.65$, $p = 0.51$). A potential explanation for this pattern of results is that expectations cannot affect happiness when participants do not have strong predictions, as it is the case immediately after reversals in the volatile condition. This would be consistent with findings from the animal literature showing that dopamine early in training does not represent prediction errors (*Coddington and Dudman, 2018*).

Finally, we focused on the win-loss model. We asked whether weights from the win-loss model were positively correlated, consistent with similar but opposite impacts. We found instead a negative correlation across participants between win and loss weights (stable: Spearman's $\rho(73) = -0.56$, $p < 10^{-6}$; volatile: Spearman's $\rho(73) = -0.68$, $p < 10^{-20}$), suggesting that individuals that respond to wins tend to respond less to losses and vice versa. Indeed, comparing the weight of wins and losses shows that participants reacted more strongly on average to losses than to wins (difference in stable: $0.69 \pm 0.12$, $z = 5.2$, $p < 10^{-6}$; difference in volatile: $0.31 \pm 0.13$, $z = 2.7$, $p < 0.01$; see *Figure 3—figure supplement 1*). Given that participants received positive feedback on average in 81% of trials in the stable environment and 63% of trials in the volatile environment, asymmetric responses to wins and losses are consistent with happiness reflecting knowledge of the underlying structure of both environments. Interestingly, the difference between win and loss weights was not correlated across participants with overall performance including the percentage of trials with positive feedback (stable: Spearman's $\rho(73) = -0.05$, $p = 0.65$; volatile: Spearman's $\rho(73) = -0.18$, $p = 0.12$) or the percentage of trials where the higher expected value option was chosen (stable: Spearman's $\rho(73) = 0.04$, $p = 0.74$; volatile: Spearman's $\rho(73) = -0.17$, $p = 0.14$).

Our results suggest that although reward information influences choice, contrary to what would be predicted from the literature, RPEs and reward magnitudes do not explain happiness when this information is not necessary for participants to learn the structure of the environment. Happiness reflects knowledge of the underlying structure of the environment in a way that cannot be explained by simple performance metrics. RPEs are relevant to learning in many paradigms, and happiness should relate to RPEs in such tasks because of their value for learning the structure of environment. Learning and reward are dissociable in our paradigm, and we find in this context that RPEs and reward magnitudes do not explain happiness.

## Sensitivity of mood dynamics to learning variables depends on volatility but not on learning rate

We found that the learning rate (i.e., behavioural sensitivity to PPE) was approximately three times higher in the volatile compared to the stable environment. Furthermore, mood dynamics were highly sensitive to PPE. However, PPE weights were actually higher in stable than volatile environments ($\Delta w_{PPE} = 0.18 \pm 0.05$, $z = 3.5$, $p < 0.001$; see *Figure 3C*). Futhermore, the difference between $w_{PPE}$ and $w_{P^-}$ was greater in stable than volatile environments (stable – volatile: $0.39 \pm 0.11$, $z = 3.7$, $p < 0.001$), consistent with a greater influence of trial-by-trial probability estimates on happiness in stable environments (see previous section). PPE weights were not correlated between participants with the learning rate in the stable (Spearman's $\rho(73) = -0.10$, $p = 0.40$) and volatile (Spearman's $\rho(73) = -0.1$, $p = 0.37$) environments nor was the difference of PPE weights across environments related to the difference in learning rate (Spearman's $\rho(73) = -0.02$, $p = 0.84$). Instead, PPE weights were highly consistent across environments (Spearman's $\rho(73) = 0.44$, $p < 0.001$, see *Figure 5A*).

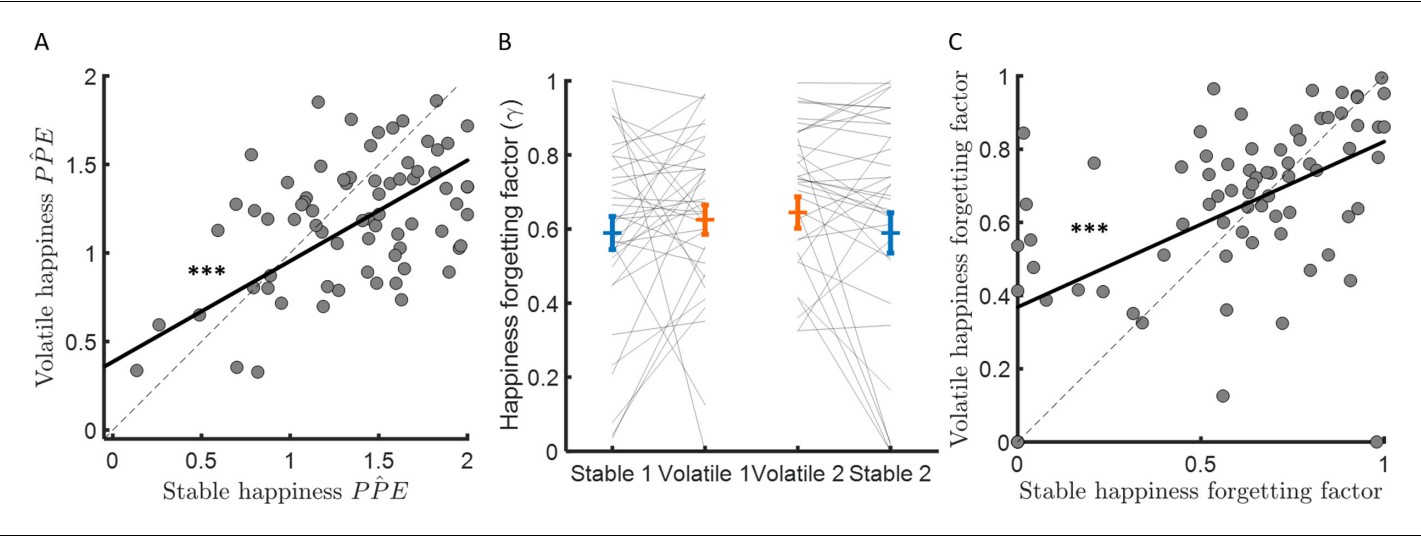

**Figure 5.** Forgetting factors are consistent across stable and volatile learning tasks. (**A**) Weights for PPEs in determining happiness were consistent across environments. (**B**) The happiness forgetting factor did not change between stable (blue) and volatile (orange) environments, regardless of testing order. See *Figure 5—figure supplement 1* for an analysis without any assumption regarding the shape of the influence decay. (**C**) Happiness forgetting factors were consistent across environments. Error bars represent SEM. ***p < 0.001.

The online version of this article includes the following figure supplement(s) for figure 5:

**Figure supplement 1.** Happiness is influenced by multiple past probability prediction errors.

The happiness model forgetting factor $\gamma$ determines how many previous trials influence current affective state. When $\gamma$ is equal to 1, mood is equally influenced by all previous trials, when $\gamma$ is equal to 0, mood is influenced by only the most recent trial. If the change in forgetting factor mirrors behaviour, forgetting factors should be lower in volatile than stable environments, reflecting integration over fewer trials and consistent with the higher learning rates observed in volatile compared to stable environments. Instead, the forgetting factor was slightly higher on average in the volatile environment (stable: $\gamma = 0.59 \pm 0.04$, volatile $\gamma = 0.63 \pm 0.03$, corresponding to current happiness being influenced by 6–7 previous trials on average in both environments, stable – volatile: $\Delta\gamma = 0.05 \pm 0.03$, z = 1.9, p = 0.064; see *Figure 5B*). Higher values for $\gamma$ in volatile environments are not consistent with happiness integrating over fewer trials as behaviour would predict. Furthermore, the change in happiness forgetting factor was not correlated across participants with the learning rate difference between environments (Spearman's $\rho(73) = -0.08$, p = 0.50). A linear regression with ten previous probability prediction errors as independent variables confirmed this model-based result (see *Figure 5—figure supplement 1*). To further test for a relationship between the forgetting factor and the learning rate, we switched the learning rates estimated from stable and volatile conditions in each individual before re-fitting happiness data (i.e., we used the 'wrong' learning rate to estimate probabilities and PPEs before fitting the happiness model and estimating a forgetting factor). This did not substantially affect estimates of the happiness forgetting factor (stable $\gamma = 0.58 \pm 0.03$, volatile $\gamma = 0.62 \pm 0.03$, stable – volatile: $\Delta\gamma = 0.032 \pm 0.026$, z = 1.4, p = 0.16). The resulting forgetting factor estimates were highly correlated with forgetting factors estimated using the actual learning rates (stable: Spearman's $\rho(73) = 0.63$, p $< 10^{-8}$; volatile: Spearman's $\rho(73) = 0.56$, p $< 10^{-6}$). The happiness forgetting factor was highly consistent across environments (Spearman's $\rho(73) = 0.41$, p < 0.001, see *Figure 5C*), suggesting that the number of previous trials that affective state depends on may be a trait-like feature of individuals unrelated to environmental volatility.

## Depressive symptoms are associated with reduced happiness in volatile environments

Previous studies have linked learning rates to anxiety (*Browning et al., 2015*; *Pulcu and Browning, 2019*) and individuals with high trait anxiety showed less ability to appropriately adjust updating of

outcome expectancies between stable and volatile environments. We found that depressive symptoms (PHQ) were uncorrelated across participants with choice accuracy (stable: Spearman's $\rho(73) = -0.06$, $p = 0.63$; volatile: Spearman's $\rho(73) = -0.09$, $p = 0.46$; volatile – stable: Spearman's $\rho(73) = 0.04$, $p = 0.71$) and all parameters estimated in the additive choice model ($\alpha$, stable: Spearman's $\rho(73) = 0.08$, $p = 0.51$; volatile: Spearman's $\rho(73) = 0.10$, $p = 0.38$; volatile – stable: Spearman's $\rho(73) = 0.19$, $p = 0.1$; $\phi$, stable: Spearman's $\rho(73) = 0.20$, $p = 0.09$; volatile: Spearman's $\rho(73) = 0.01$, $p = 0.94$; volatile – stable: Spearman's $\rho(73) = -0.20$, $p = 0.09$; $\beta$, stable: Spearman's $\rho(73) = -0.04$, $p = 0.76$; volatile: Spearman's $\rho(73) = -0.07$, $p = 0.53$; volatile – stable: Spearman's $\rho(73) = 0.04$, $p = 0.75$). In the stable environment, where uncertainty and volatility are low, average happiness did not correlate across participants with depressive symptoms (Spearman's $\rho(73) = 0.07$, $p = 0.58$; *Figure 6A*, left panel). In the volatile environment, where uncertainty is high and volatility is high, average happiness was correlated with depressive symptoms, with lower happiness associated with higher depressive symptoms (Spearman's $\rho(73) = -0.23$, $p = 0.043$; *Figure 6A*, central panel). Finally, the difference between average happiness between volatile and stable environments was also correlated with depressive symptoms even after standardising the variables (*Wilcox and Tian, 2008*) (volatile – stable: Spearman's $\rho(73) = -0.28$, $p = 0.014$; *Figure 6A*, right panel). Baseline

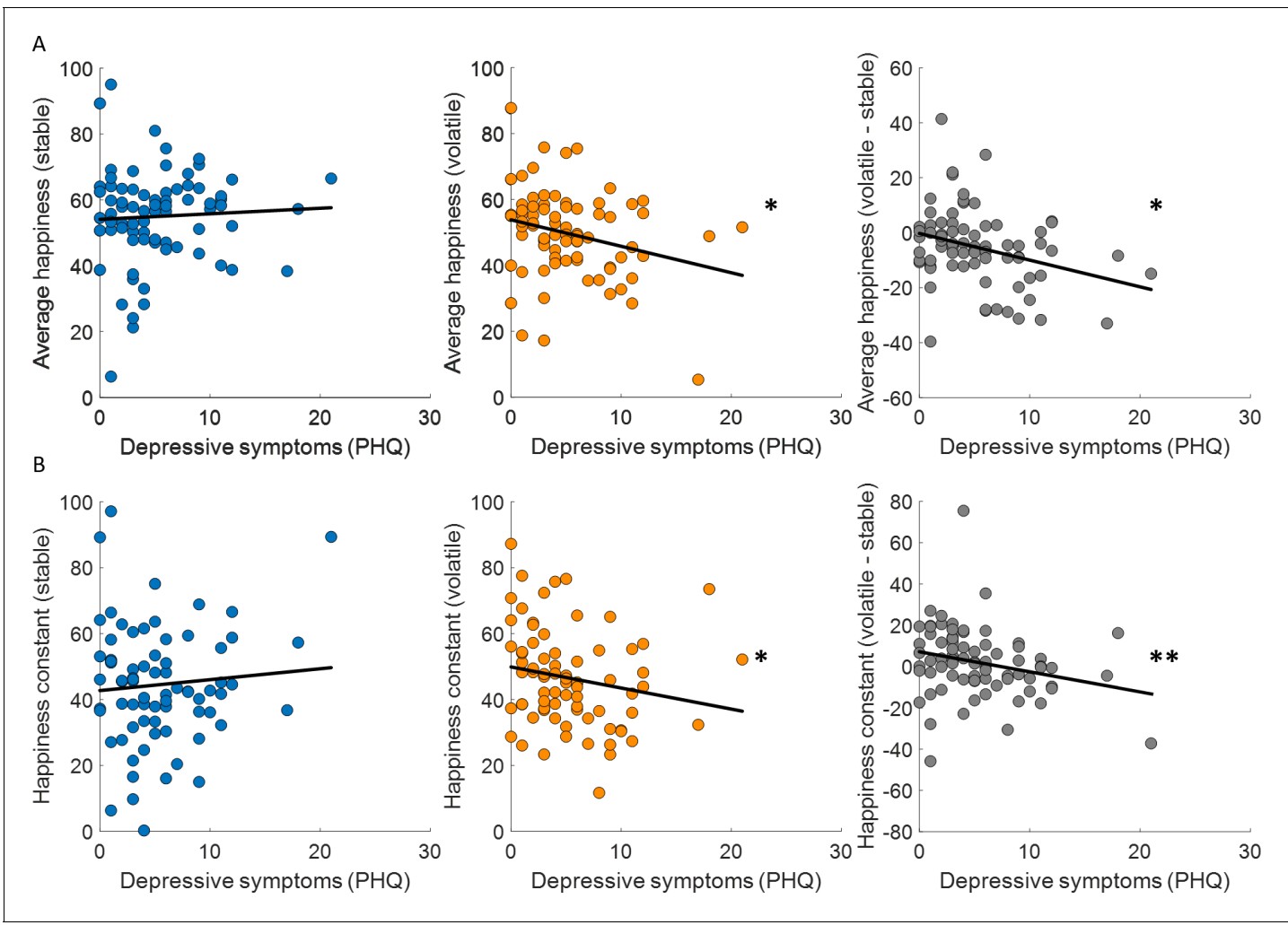

**Figure 6.** Baseline mood decreases with depressive symptoms in volatile environments. (**A**) Average happiness was not correlated with depressive symptoms (PHQ) in the stable task (left panel, blue) but decreased with depressive symptoms in the volatile task (middle panel, orange). The difference in happiness between stable and volatile environments was also significantly related to depression (right panel). (**B**) Baseline mood parameters estimated with non-z-scored happiness ratings showed the same relationship to depressive symptoms as average happiness with lower parameters in volatile than stable environments. *p < 0.05, **p < 0.01.

mood parameters estimated using our happiness model fit to non-z-scored happiness ratings showed the same relationship to depressive symptoms (stable: Spearman's $\rho(73) = -0.07$, p = 0.54; volatile: Spearman's $\rho(73) = -0.28$, p = 0.017; volatile – stable, standardised: Spearman's $\rho(73) = -0.32$, p = 0.0049; see *Figure 6B*).

No other happiness model parameters were correlated with depressive symptoms ($w_P$; stable: Spearman's $\rho(73) = 0.20$, p = 0.09; volatile: Spearman's $\rho(73) = 0.15$, p = 0.19; volatile – stable: Spearman's $\rho(73) = -0.02$, p = 0.84; $w_{PPE}$, stable: Spearman's $\rho(73) = 0.09$, p = 0.45; volatile: Spearman's $\rho(73) = 0.07$, p = 0.54; volatile – stable: Spearman's $\rho(73) = -0.03$, p = 0.81; $\gamma$, stable: Spearman's $\rho(73) = 0.06$, p = 0.63; volatile: Spearman's $\rho(73) = -0.09$, p = 0.44; volatile – stable: Spearman's $\rho(73) = -0.12$, p = 0.28).

## Discussion

We found that subjects tracked outcome probabilities and made decisions by integrating both learned probability and explicit reward magnitudes. Learning rates adapted to environmental volatility, with a higher learning rate in the more volatile environment consistent with previous studies (*Behrens et al., 2007*; *Browning et al., 2015*; *Massi et al., 2018*). That behaviour was consistent with an additive choice model (*Donahue and Lee, 2015*; *Farashahi et al., 2017*; *Massi et al., 2018*; *Rouault et al., 2019*) which is consistent with recent empirical evidence (*Farashahi et al., 2019*; *Koechlin, 2020*) that humans and non-human primates adopt a multiplicative strategy under risk when probabilities are explicit, but both species adopt an additive strategy under uncertainty when probabilities must be learned.

Our tasks required learning the probability of getting a reward and the reward magnitude was explicitly given. In such an environment, mood dynamics were more closely related to learning-relevant variables than learning-irrelevant variables and we found convergent evidence that this was the case across both stable and volatile learning tasks. Mood was sensitive to the combined influence of past chosen subjective probabilities and past PPEs. Parameters for PPE and forgetting factors estimated from happiness ratings were correlated across stable and volatile environments. Finally, we found that although choice accuracy and choice model parameters were not affected by depressive symptoms when changes between safe and volatile environments are signalled, the decrease in happiness observed in the volatile relative to the stable environment was correlated with symptom severity. The same pattern was present for the baseline mood parameter in the happiness model. Experiencing a stable environment with low uncertainty and volatility could attenuate the expression of depressive symptoms on mood. Risky decision tasks used in previous studies (*Rutledge et al., 2017*; *Rutledge et al., 2015*; *Rutledge et al., 2014*) maximise irreducible uncertainty (i.e., risky options had a 50% probability of each option), which is more comparable to the volatile environment in the current study and may explain the previous finding of a link between baseline mood parameters and depressive symptoms. Computational models that capture ecologically relevant learning and decision processes may provide a critical advantage for understanding the mechanisms that underlie psychiatric symptoms (*Scholl and Klein-Flügge, 2018*). Our findings suggest that subjective feelings measured during tasks in depression-relevant domains may provide additional information not captured by computational models of learning and decision-making. One reason depression might reduce mood more in volatile than stable environments could be an increase in the number of negative prediction errors experienced. Misestimation of the level of uncertainty may also lead to a tendency for negative events to disproportionally affect depressed individuals, and this uncertainty misestimation is believed to contribute to depression and anxiety (*Pulcu and Browning, 2019*). That the learning rate difference between the volatile and the stable environment did not correlate with anxiety symptom severity is consistent with previous findings (*Browning et al., 2015*) that anxiety is linked to learning deficits for aversive but not appetitive outcomes. In the aversive domain, anxiety severity might be associated with differences in behavioural adaptation to volatility changes as well as mood.

It is not yet established what the neural signal associated with PPEs is. On the one hand, reward prediction errors have been associated with neuromodulator dopamine and are thought to be linked to ventral tegmental area (*Bayer and Glimcher, 2005*; *Cohen et al., 2012*; *Hart et al., 2014*; *Montague et al., 1996*; *Pessiglione et al., 2006*; *Schultz et al., 1997*). In most studies, RPEs and PPEs are equivalent and therefore the specific link between PPEs and dopamine is less documented.

The probability of obtaining reward and probability prediction error have been associated with VTA activity correcting for expected value (*Behrens et al., 2007*), suggesting that PPEs may be represented by dopamine. Boosting dopamine levels pharmacologically during risky decision making increases the happiness resulting from smaller rewards to a level similar to that resulting from larger rewards (*Rutledge et al., 2015*). Although happiness in that study was influenced by the history of RPEs, dopamine drug impacts were limited to rewards. Dopamine has also been associated with other signals than prediction errors, for example incentive salience which might relate to 'wanting' and might influence choice and action (*Berridge, 2012*; *Smith et al., 2011*; *Zhang et al., 2009*).

Some studies suggest that dopaminergic activity in the midbrain is linked to information-seeking to reduce uncertainty about an upcoming reward, even though such information is not instrumental (*Bromberg-Martin and Hikosaka, 2009*; *Brydevall et al., 2018*; *Charpentier et al., 2018*; *Gruber and Ranganath, 2019*). The intrinsic reward resulting from reducing uncertainty could influence mood, and mood ratings could then be used to quantify the relative subjective weight of extrinsic and intrinsic reward.

Previous studies using risky decision tasks where reward and probability were explicitly represented (*Rutledge et al., 2014*; *Rutledge et al., 2015*) showed that mood dynamics were explained by past expected values and RPEs. The present design allows us to dissociate the impact of learning-relevant and learning-irrelevant information for mood in two different standard learning environments. Our results suggest that when goal attainment requires adaptive behaviour, mood dynamics reflect learning-relevant information. Consistent with the results obtained from risky decision tasks used in previous studies (*Rutledge et al., 2014*; *Rutledge et al., 2015*; *Rutledge et al., 2017*), potential rewards were a key determinant of behaviour in our task. However, rewards were not a determinant of mood in the current study, in contrast to previous results in risky decision tasks (*Rutledge et al., 2014*; *Rutledge et al., 2015*; *Rutledge et al., 2017*). Unlike most reinforcement learning experiments, our task design allows dissociating the impacts of PPEs and RPEs on behaviour and mood. Here, our results suggest that happiness does not always depend on reward and preferentially reflects learning about the structure of the environment. However, if learning the structure of the environment requires tracking changing reward magnitudes, we would expect that happiness would track learning-relevant variables (e.g., reward magnitudes and RPEs in such an environment).

This result might imply a role for mood in learning in line with influential proposals (*Eldar et al., 2016*; *Eldar and Niv, 2015*). However, mood did not reflect all learning-relevant information that influenced behaviour. Happiness forgetting factors corresponding to the number of past trials that influence affective state were highly correlated across environments and acted more as a stable trait that differed between individuals and did not adjust to environmental volatility. Overall, our findings show that mood dynamics are sensitive to depressive symptoms and reflect variables relevant to adaptive behaviour irrespective of environmental volatility.

## Materials and methods

### Subjects

Seventy-five healthy subjects (age range 18–35, 24 males) took part in the experiment. Thirty-seven completed the stable learning task first and the volatile learning task second. Group allocation was randomised. Subjects were paid £10 for their participation. The number of participants recruited for the current cohort was selected to provide >95% power of detecting a similar effect size as that reported in a previous study in which a volatility manipulation was used to influence learning rate (*Browning et al., 2015*). All subjects gave informed consent and the Research Ethics Committee of University College London approved the study (Committee approval ID Number: 12673/001).

### Procedure

Participants were first instructed about the tasks. They performed 20 practice trials before a test ensuring that they understood that both probability and magnitude mattered to maximise the number of points obtained. They were told that they would be exposed to two environments: an environment where one car is more likely to win for the entire session, and an environment where which car is more likely to win changes occasionally. Because we wanted to maximise the efficiency of the behavioural manipulation (i.e., the difference in learning rate between environments observed in

previous studies [*Behrens et al., 2007*; *Browning et al., 2015*; *Massi et al., 2018*]) to study how mood dynamics varied with behavioural sensitivity, we explicitly signalled the environment by using different pairs of cars in the two environments. Moreover, before each condition, an instruction screen explained in which environment participants will be placed. Finally, a fixation symbol displayed in the centre of the screen in each trial was specific for each environment: '-' for the stable environment and '~' for the volatile environment. Participants were given no guidance as to how they should use information about environmental volatility. After completion of the task, participants completed three standard clinical questionnaires: Beck Depression Inventory (BDI-II *Beck et al., 1996*), Patient Health Questionnaire (PHQ-9, *Kroenke et al., 2001*), and the State/Trait Anxiety Inventory (STAI, *Spielberger, 1983*).

## Experimental task

The task was implemented using the Cogent toolbox in MATLAB (MathWorks, Inc). Subjects had to choose between two cars, each associated with a probability (20% or 80%) of winning. If the chosen car won, participants earned the corresponding amount of points. In the stable environment (80 trials), the probability to win for the best car was 80%. In the volatile environment (80 trials), reward probabilities switched between 80% for one car and 80% for the other car every 20 trials. The order was counterbalanced between subjects (n = 38 in stable-volatile order, n = 37 in volatile-stable order). The outcomes were locally pseudo-randomised, to ensure that every 10 trials (i.e., trials 1–10, 11–20), the car with the highest outcome probability won on exactly 8 of 10 trials. The possible pairs of rewards were 10–10, 10–40, 10–60, 10–80, 20–40, 40–10, 40–20, 40–40, 60–10, 80–10. Subjects were primed when the side of the screen for each car was swapped (every six to ten trials) with an explicit cue. They were also instructed that the car location was unrelated to the outcome probability and to the change in outcome probabilities in the volatile environment. Every three to four trials, subjects were asked to indicate 'How happy are you right now?' on a scale from very unhappy to very happy. They were told to consider these extremes within the context of the experiment. Each trial started with a fixation screen for a duration varying between 0.9 and 1.9 s. Cars were displayed for 1.2 s without any information about reward magnitudes, and no choice was allowed in this phase. Subjects were free to choose the option they preferred without any time constraints as soon as the potential reward for each car was displayed. The chosen option was surrounded by a yellow frame for 1.5 s. Finally, the outcome was displayed for 2 s. Both the car and the reward magnitude frames were green if the chosen car won. They were red and the car was crossed out if the chosen car lost.

## Data analyses

Two-sided Wilcoxon signed rank tests were used to compare performance, proportion of win-stay/lose-shift, and model parameters between environments at the group level. Spearman rank correlations across participants were used to test for relationships between parameters and to relate depression scores to behavioural and happiness parameters. All analyses were performed using MATLAB. To test whether the correlation between PHQ score and happiness baseline parameters was higher in the volatile condition than in the stable condition, we correlated the standardised difference between the happiness baseline parameter in the stable and volatile conditions with the standardised PHQ score which quantifies depressive symptoms (*Wilcox and Tian, 2008*). The analysis codes were written in MATLAB and are available at Github (https://github.com/BastienBlain/MSWB_LearningNotReward; copy archived at swh:1:rev:b7c4a0cd761dcf249c72caf809dd81af24c4a49b; *Blain, 2020*).

## Computational models

All models were fitted to experimental data by minimising the negative log likelihood of the predicted choice probability given different model parameters using the fmincon function in MATLAB (Mathworks Inc). More specifically, parameters were treated as random effects that could differ between subjects (*Kreft and De Leeuw, 1998*): data were fitted for each participant and statistical tests were performed at the group level. We used standard model comparison techniques (*Burnham and Anderson, 2004*; *Schwarz, 1978*) to compare model fits. For each model fit in individual subjects, we computed the Bayesian Information Criterion (BIC), which penalises for model complexity (i.e. number of parameters), and then summed BIC across subjects. The model with the

lowest BIC is the preferred model. For all choice models, the learning rate was bounded between 0 and 1, the inverse temperature between 0 and 50 (to avoid ceiling effects), the probability-magnitude relative weight phi between 0 and 1, and the gamma risk aversion parameters between 0 and 10. Note that reward magnitude was normalised between 0 and 1 in the additive model. For the happiness models, we first fitted each happiness model on both environments using the same parameters for both environments. Then each model was fitted for each environment separately, with the starting parameters determined by the joint model fit under standard constraints (the forgetting factor could vary only between 0 and 1 and the baseline mood parameter could only vary between 0 and 100). Because participants vary in how they use the scale, we z-scored happiness ratings for all the analyses reported in the main text, except in the analyses where we asked how the baseline mood parameters are related to depression.

### Estimated frequency and exceedance probability

Models were treated as random effects that could differ between subjects and have a fixed (unknown) distribution in the population. Model frequency with which any model prevails in the population, as well as exceedance probability (EP), which measures how likely it is that any given model is more frequent than all other models in the comparison set (*Stephan et al., 2009*), were estimated using the VBA Matlab toolbox (*Daunizeau et al., 2014*). See figure supplementary figure 2 for an illustration of the estimated frequency for three different model spaces. An EP greater than 0.95 is considered significant.

### Happiness is influenced by multiple past probability prediction errors

To estimate the influence of the past trials on the current happiness without any assumption regarding the shape of the influence decay, we fitted a general linear model including for each rating the previous 10 probability prediction errors using the Matlab glmfit function. Each value was then tested against 0 at the group level using two-sided Wilcoxon signed rank tests (see supplementary figure 3).

## Acknowledgements

We thank Sankalp Garud, Matilde Vaghi, Benjamin Chew, Paul Sharp, and Rachel Bedder for their help. R.B.R. is supported by a Medical Research Council Career Development Award (MR/N02401X/1), a 2018 NARSAD Young Investigator Grant (27674) from the Brain and Behavior Research Foundation, P and S Fund, and by the National Institute of Mental Health (1R01MH124110). The Max Planck UCL Centre is a joint initiative supported by UCL and the Max Planck Society. The Wellcome Centre for Human Neuroimaging is supported by core funding from the Wellcome Trust (203147/Z/16/Z).

## Additional information

### Funding

| Funder | Grant reference number | Author |
|---|---|---|
| Medical Research Council | MR/N02401X/1 | Robb B Rutledge |
| Brain and Behavior Research Foundation | Young Investigator Grant (27674) | Robb B Rutledge |
| NIMH | 1R01MH124110 | Robb B Rutledge |
| Max Planck Society | Centre Grant | Robb B Rutledge |
| Wellcome Trust | 203147/Z/16/Z | Robb B Rutledge |

The funders had no role in study design, data collection and interpretation, or the decision to submit the work for publication.

### Author contributions

Bastien Blain, Conceptualization, Data curation, Formal analysis, Validation, Investigation, Visualization, Methodology, Writing - original draft, Writing - review and editing; Robb B Rutledge,

Conceptualization, Resources, Formal analysis, Supervision, Validation, Investigation, Project administration, Writing - review and editing

### Author ORCIDs
Bastien Blain ⬩ https://orcid.org/0000-0002-7735-6043
Robb B Rutledge ⬩ https://orcid.org/0000-0001-7337-5039

### Ethics
Human subjects: All subjects gave informed consent and the Research Ethics Committee of University College London approved the study study (Committee approval ID Number: 12673/001).

### Decision letter and Author response
Decision letter https://doi.org/10.7554/eLife.57977.sa1
Author response https://doi.org/10.7554/eLife.57977.sa2

# Additional files

## Supplementary files
• Transparent reporting form

## Data availability
Data and code are available online (https://github.com/BastienBlain/MSWB_LearningNotReward; copy archived at https://archive.softwareheritage.org/swh:1:rev:b7c4a0cd761dcf249c72caf809d-d81af24c4a49b/).

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
