## [Decision Letter]

**Acceptance summary:**

It has been previously shown that happiness or subjective well-being is related to the recency-weighted integration of reward prediction errors (RPE), rather than the magnitude of reward. However, whether reward magnitude or probability plays a more important role in determining happiness had not been tested. In the present study, the authors show that happiness was more reliably related to the probability prediction error (PPE) than RPE, and therefore a factor more important for learning than reward magnitude.

**Decision letter after peer review:**

Thank you for submitting your article "Mood dynamics depend on learning and not reward" for consideration by *eLife*. Your article has been reviewed by three peer reviewers, including Daeyeol Lee as the Reviewing Editor and Reviewer #1, and the evaluation has been overseen by Joshua Gold as the Senior Editor.

The reviewers have discussed the reviews with one another and the Reviewing Editor has drafted this decision to help you prepare a revised submission.

Summary:

It has been shown previously that the subjective well-being (SWB, happiness) is correlated with the recency-weighted reward-prediction errors (RPE). In this manuscript, the authors showed that during a probabilistic learning task, the SWB is better accounted for by a set of recent probability prediction errors (PPE) than by reward prediction errors (RPE). The key goals of this paper were to (1) test additive and multiplicative models of choice, (2) assess the relationship of learning rate and volatility, and (3) and understand what governs "happiness" during choice and learning. Although RPE is more directly related to the income of the subject and also more closely related to the expected value that is presumably used during decision making, it is PPE that's relevant for learning. Therefore, this finding implies that the subjective mood or SWB might be more related to learning, rather than how much the desirability of the decision outcome deviated from the expected outcome. The experiments and analyses are carefully conducted, and the results are presented clearly.

Essential revisions:

1) The finding that probability (errors) but not RPEs mediate (or are related to) self-reports of happiness might be limited to the specific study design used. Imagine you enter a reward rich patch (ice cream) and you eat – your happiness may rise somewhat exponentially as a function of the amount of ice cream you can eat (especially if you think someone may come and take it away; what amount you "got away with" is probably a pretty good happiness indicator and is a dynamic-RPE related variable). Another example is one related to gambling. Imagine you make a gamble and loose. Depending on your state, context, and history, either RPEs or PPEs could be related to your "happiness". If the authors agree, then it might be helpful to temper the claims and stating the limitations (e.g. how much do these results inform other contexts).

2) It might be helpful if the authors include a couple of additional models (in addition to models given by Equations 10 and 11) to examine whether SWB is related to variables other than the ones already tested in this study. For example, although this might have been shown in previous studies, the authors have another opportunity to test how strongly the SWB is related to the recency-weighted reward magnitude. In addition, how poorly do the models like this perform if they include the recency-weighted RPE, rather than PPE (e.g., EV deviations and RPE).

3) The authors justify concentrating on additive models of happiness based on their own previous work. This is not entirely convincing in this study, however. Equations 6-7 of happiness are additive. Have they tested alternative models? Also, related to model conceptualization, for happiness model with P + PPE, isn't this basically just counting wins – assuming forgetting factors are the same? Is this consistent with psychological work on happiness and mood modulation?

4) Reward-related variables are sometimes referred as "choice-relevant variables", but its rationale should be explained more clearly. Why aren't the variables relevant for learning not relevant for choice?

5) When predicting happiness ratings, did authors use a random-effects or fixed-effect model? The relevant details are missing. If a random-effects model was not used, this should be used (instead of a fixed-effects model).

6) Authors claimed that "depressive symptoms reduce happiness more in volatile than stable environments” (Abstract), but it is not clear whether this was tested statistically in the manuscript. Authors seem to have shown that average happiness significantly correlated with depressive symptoms only in the volatile environment. Authors should perform a direct and statistical comparison to support the conclusion (Niewenhius et al., 2011; Wilcox and Tian, 2008).

---

## [Author Response]

Essential revisions:1) The finding that probability (errors) but not RPEs mediate (or are related to) self-reports of happiness might be limited to the specific study design used. Imagine you enter a reward rich patch (ice cream) and you eat – your happiness may rise somewhat exponentially as a function of the amount of ice cream you can eat (especially if you think someone may come and take it away; what amount you "got away with" is probably a pretty good happiness indicator and is a dynamic-RPE related variable). Another example is one related to gambling. Imagine you make a gamble and loose. Depending on your state, context, and history, either RPEs or PPEs could be related to your "happiness". If the authors agree, then it might be helpful to temper the claims and stating the limitations (e.g. how much do these results inform other contexts).

We agree with the reviewers that our finding that happiness preferentially depends on learning and not necessarily on reward may be related to specific features of the task design. In this case, we think the key relevant feature is that computing RPEs is not adaptive for learning in this environment because PPEs are the quantity required for learning. We agree with the reviewers that the variables that determine happiness should depend on state, context, and history and these are all important areas for future research. We have tempered our claims and stated the limitations in several sections. We have revised the text in the Results as follows:

“Our results suggest that although reward information influences choice, contrary to what would be predicted from the literature, RPEs and reward magnitudes do not explain happiness when this information is not necessary for participants to learn the structure of the environment. […] Learning and reward are dissociable in our paradigm, and we find in this context that RPEs and reward magnitudes do not explain happiness.”

We have revised the relevant text in the Discussion as follows:

“Previous studies using risky decision tasks where reward and probability were explicitly represented (Rutledge et al., 2014, 2015) showed that mood dynamics were explained by past expected values and RPEs. […] However, if learning the structure of the environment requires tracking changing reward magnitudes, we would expect that happiness would track learning-relevant variables (e.g., reward magnitudes and RPEs in such an environment).”

2) It might be helpful if the authors include a couple of additional models (in addition to models given by Equations 10 and 11) to examine whether SWB is related to variables other than the ones already tested in this study. For example, although this might have been shown in previous studies, the authors have another opportunity to test how strongly the SWB is related to the recency-weighted reward magnitude. In addition, how poorly do the models like this perform if they include the recency-weighted RPE, rather than PPE (e.g., EV deviations and RPE).

We previously reported in risky decision tasks (Rutledge et al., 2014, 2015) that recent reward magnitudes are not as good a predictor of SWB as recent RPEs. However, we agree that a thorough test of this possibility for our data set would be valuable, especially because this is the first study to describe SWB in a reinforcement learning context. We included the models described below in the Results. Interestingly, the win-loss model (which also does not incorporate reward magnitude) suggested by the reviewers in major point 3 has a similar overall BIC to the P^+PPE^ model. Our findings related to the win-loss model strengthens our understanding of the differences between happiness in stable and volatile environments and we discuss these findings at length in response to major point 3 and in the revised manuscript. Briefly, happiness preferentially reflects learning the structure of the world and not reward magnitudes in both stable and volatile environments but not in a way that depends on model-derived expectations to the same degree. To provide a thorough test of models that incorporate reward magnitudes, the Results now include additional models and we have revised the text as follows:

“We next extended the model space with plausible alternative models. We included two models incorporating the history of reward magnitude. […] Comparison of weights across tasks therefore suggests a reduced impact of expectations on happiness as environmental volatility increases.”

The updated Table 2 includes information comparing all models tested.

3) The authors justify concentrating on additive models of happiness based on their own previous work. This is not entirely convincing in this study, however. Equations 6-7 of happiness are additive. Have they tested alternative models?

There are two terms in Equations 6 and 7: a constant term and the prediction error term (PPE and RPE, respectively). Our simple additive models already account for almost half of the variance in happiness ratings (mean r^2^ = 0.36-0.50). However, because probabilities for consecutive trials are not independent, interaction terms could be important in reinforcement learning tasks for understanding how happiness changes over time.

We tested whether there was a non-linear effect of recent PPEs on happiness. To that end, we added to the PPE model a “boost” term when two consecutive PPEs have the same sign: two consecutive positive PPEs boost happiness (encoded as 1), whereas two consecutive negative PPEs decrease happiness (encoded as -1). This is in addition to the linear effects in our original model.

Happiness(t)=w0+wPPE^∑j=1tγt−jPPEj^+wboost∑j=1tγt−jBoostj Positive weights correspond to an enhanced PPE impact when the PPE has the same sign as the PPE on the previous trial. A negative weight corresponds to a reduced PPE impact when the sign of consecutive PPEs is the same. This boost weight was not significantly different from 0 in both the stable (0.19 ± 0.11, z = 0.40, p = 0.68) and volatile (0.21 ± 0.12, z = 0.60, p = 0.55) environments. Model comparison shows that including this additional term is not supported by the data.

**Table resptable1:** 

PPE^	3	0.48	0.42	-640	-436	242	710
PPE^ + boost	4	0.53	0.47	-595	-408	287	737

Also, related to model conceptualization, for happiness model with P + PPE, isn't this basically just counting wins – assuming forgetting factors are the same? Is this consistent with psychological work on happiness and mood modulation?

The effect of expectations on happiness have been reported in multiple studies (Rutledge et al., 2014, 2015). We have revised the Introduction to report some additional psychological studies that find consistent results across several labs:

“Emotions are widely believed to play a role in adaptive behaviour (Fredrickson, 2004), but no computational framework exists to link them. […] Mood has been proposed to represent environmental momentum, whether an environment is getting better or worse, which could be a useful variable for adaptive behaviour (Eldar et al., 2016; Eldar and Niv, 2015).”

Evaluating an alternative win-loss model that tests whether participants are, in a sense, counting wins and losses led to some intriguing results. We believe that these results provide a more complete picture of the relationship between happiness and reinforcement learning, particularly in volatile environments, and we appreciate the suggestion. We have added the following text to the Results:

“We also asked whether the history of wins (excluding any information about reward magnitude) and losses could account for happiness by fitting the following model:(16)Happiness(t)=w0+wwin∑j=1tγt−jwinj−wloss∑j=1tγt−jlossj

[…] Learning and reward are dissociable in our paradigm, and we find in this context that RPEs and reward magnitudes do not explain happiness.”

4) Reward-related variables are sometimes referred as "choice-relevant variables", but its rationale should be explained more clearly. Why aren't the variables relevant for learning not relevant for choice?

This is correct. The variables relevant for learning are also relevant for choice but the converse is not correct. We aimed to find a dissociation between learning-relevant variables, which are indeed necessary to make the best decisions to maximise income, and learning-irrelevant variables that are not relevant to learning but are relevant to make the best decisions to maximise income. In particular, the RPE is a learning-irrelevant variable because computation of this quantity is not part of the learning models that describe behaviour. Similarly, EV is a learning-irrelevant variable because, although it is related to choice, it is not part of the learning process.

We agree that the term ‘choice relevant’ is inaccurate and misleading, and we have replaced the term with ‘learning irrelevant’.

5) When predicting happiness ratings, did authors use a random-effects or fixed-effect model? The relevant details are missing. If a random-effects model was not used, this should be used (instead of a fixed-effects model).

We agree with the reviewers that a fixed-effects model is not appropriate for these analyses. Details have been added to the Materials and methods to explain our model-fitting procedure more accurately, which corresponds to using random-effects models:

“All models were fitted to experimental data by minimizing the negative log likelihood of the predicted choice probability given different model parameters using the fmincon function in MATLAB (Mathworks Inc). […] We used standard model comparison techniques (Burnham and Anderson, 2004; Schwarz, 1978) to compare model fits.”

6) Authors claimed that "depressive symptoms reduce happiness more in volatile than stable environments” (in Abstract), but it is not clear whether this was tested statistically in the manuscript. Authors seem to have shown that average happiness significantly correlated with depressive symptoms only in the volatile environment. Authors should perform a direct and statistical comparison to support the conclusion (Niewenhius et al., 2011; Wilcox and Tian, 2008).

Originally, we tested for this difference by correlating the difference in the happiness baseline parameter between the stable and volatile conditions with PHQ score which quantifies depressive symptoms, without standardising. This relationship was shown in Figure 6 (right panel) and was significant (Spearman’s ρ(73) = -0.32, p < 0.01). However, according to the more conservative approach described in one of papers mentioned, Wilcox and Tian, 2008, the variables should be standardised. Standardising the variables also leads to a negative correlation between the difference between volatile and stable happiness parameters and depressive symptoms (Spearman’s ρ(73) = -0.28, p = 0.014). Similar results are found for the baseline mood parameter after standardising the variables (Spearman’s ρ(73) = -0.32, p = 0.0049). We have added this result to the manuscript:

“In the volatile environment, where uncertainty is high and volatility is high, average happiness was correlated with depressive symptoms, with lower happiness associated with higher depressive symptoms (Spearman’s ρ(73) = -0.23, p = 0.043; Figure 6A, central panel). […] Baseline mood parameters estimated using our happiness model fit to non-z-scored happiness ratings showed the same relationship to depressive symptoms (stable: Spearman’s ρ(73) = -0.07, p = 0.54; volatile: Spearman’s ρ(73) = -0.28, p = 0.017; volatile – stable, standardised: Spearman’s ρ(73) = -0.32, p = 0.0049; see Figure 6B).”

We have explained this analysis in the revised Materials and methods:

“Two-sided Wilcoxon signed rank tests were used to compare performance, proportion of win-stay/lose-shift, and model parameters between environments at the group level. […] To test whether the correlation between PHQ score and happiness baseline parameters was higher in the volatile condition than in the stable condition, we correlated the standardised difference between the happiness baseline parameter between the stable and volatile conditions with the standardised PHQ score which quantifies depressive symptoms (Wilcox and Tian, 2008).”